# Compound Microalgae-Type Biofunctional Hydrogel for Wound Repair during Full-Thickness Skin Injuries

**DOI:** 10.3390/polym16050692

**Published:** 2024-03-03

**Authors:** Yi Mao, Yajuan Sun, Cheng Yang

**Affiliations:** Key Laboratory of Synthetic and Biological Colloids, Ministry of Education, School of Chemical and Material Engineering, Jiangnan University, Wuxi 214122, China; maoyi@stu.jiangnan.edu.cn

**Keywords:** chitosan/spirulina hydrogel, antibacterial, antioxidant, hemostatic, anti-inflammatory, wound healing

## Abstract

A dual biofunctional hydrogel (HQCS-SP) wound dressing, offering antibacterial properties and a biological response, was innovatively designed and developed to repair full-layer skin defects. The HQCS-SP hydrogel creates an artificial matrix that facilitates cell recruitment, extracellular matrix deposition, exhibiting exceptional tissue affinity, robust self-healing, effective hemostatic capabilities and accelerates wound healing. It is synthesized by crosslinking modified chitosan (HQCS) with spirulina protein (SP) and Fe^3+^. The HQCS provides antibacterial, antioxidant, good tissue affinity and excellent hemostasis performance. The incorporation of SP not only reinforces the antioxidant, antibacterial, anti-inflammatory, and pro-angiogenesis effects but also participates in the regulation of signal pathways and promotes wound healing. Therefore, this study offers a new visual angle for the design of advanced functional trauma dressings with great application potential in the bio-medical field.

## 1. Introduction

Skin is the tissue that covers the surface of the human body and wraps around the muscles, which plays an essential role in resisting the invasion of foreign substances, preventing physical and chemical damage, maintaining body fluid homeostasis, and so on [1,2,3]. Various factors such as accidents, surgeries, poor living conditions, an aging population, and diabetes lead to acute skin injuries (burns, cuts, abrasions, etc.) and chronic skin injuries (eczema, psoriasis, chronic ulcers, etc.), imposing significant burdens on patient’s families, healthcare systems, and society [4,5,6]. Usually, epidermal damage can be repaired entirely by division, proliferation, and differentiation of basal cells [7,8]. However, the repair of full-thickness damage to the dermis is complex due to the complicated trauma pathological microenvironment, including an inflammatory response, oxidative stress, and repeated bacterial infection [9,10,11].

Extensive efforts have been devoted to addressing this challenge [12,13]. Among them, hydrogel dressings have demonstrated many excellent properties, which could offer an artificial matrix in the way of wound dressing for the recruitment of cells and the deposition of extracellular matrix (ECM), while supplying structural reinforcement for cell movement and growth at the same time [14,15]. Additionally, they can absorb the secretions from a wound and maintain a damp environment in the affected area. More importantly, hydrogel with various functions (such as antibacterial, antioxidant, anti-inflammatory, and so on) could be easily attained by using different polymer frameworks or incorporating bio-actives into the hydrogel [16,17,18]. Considering all these superiorities, hydrogel may be a promising wound dressing for full-thickness damage.

Bacterial infection at the wound site is vital for full-thickness damage recovery [19]. It could induce inflammation and oxidative stress, seriously hindering wound healing [20,21]. Many strategies have been proposed to combat bacterial infections. Among them, chitosan (CS), as a famous antibacterial hydrogel material, has many unique properties, such as biodegradability, biocompatibility, and gel-forming properties, which have gained significant attention [22,23,24]. Previously, Z. Lu et al. prepared the Ag/ZnO loaded CS composite dressing to combat bacterial infections for wound healing [25]. Zhang et al. successfully prepared catechol-functionalized CS thermosensitive hydrogel, and verified that hydrogels play a vital role in inhibiting inflammatory cell aggregation and promoting wound healing in multiple stages [26]. However, the application of pure CS without any modifications was hindered due to its poor water solubility, which demonstrated limited antibacterial activity [27]. Thus, it is imperative to modify CS to improve its inherent aqueous solubility and fortified antibacterial capacity for wound healing.

In addition to bacterial infection, the prolonged and heightened inflammatory reaction further obstructs the healing of full-thickness injuries [28]. Furthermore, the reactive oxygen species (ROS) generated by inflammatory cells at the wound location exacerbate oxidative stress-related harm, leading to various detrimental consequences [29]. These include diminished collagen production and blood vessel formation, improper breakdown of the ECM and growth factors, and delayed re-epithelialization processes [30]. Spirulina, an abundant natural resource in the biosphere, is rich in protein, polysaccharides, vitamins, and minerals [31]. In particular, proteins and polysaccharides, which account for more than 90%, have supported strong cell protection, liver protection, and neuroprotective properties due to their excellent anti-inflammatory, antioxidant, and anti-cancer properties while being biodegradable in the physiological environment [32,33,34]. For example, Shiou-Chi Cherng et al. reported that spirulina protein (SP) could suppress nitrite production and expression of messenger molecules in lipopolysaccharide (LPS)-stimulated RAW 264.7 macrophages [35]. Moreover, SP has demonstrated the ability to enhance collagen synthesis in CCD-986sk cells and decrease elastase activity, which plays a significant positive role in wound repair. Furthermore, it can promote cell migration and proliferation through specific kinase pathways [36]. These functions of SP on human fibroblasts make it possible to promote skin trauma repair [37].

In this work, we envisioned a biofunctional hydrogel (HQCS-SP) for wound healing by rational design, with injectable, self-healing, antibacterial, anti-inflammatory, and antioxidant properties (Figure 1). First, quaternary ammonium groups and catechol were grafted onto the CS backbone to enhance the water solubility and tissue affinity. Secondly, natural SP was bonded with modified CS (HQCS) through hydrogen bonding. Finally, the complex hydrogel was cross-linked by complexation between HQCS with SP and ferric iron (Fe^3+^). The formed complex hydrogel demonstrated strong tissue affinity, good injectability, efficient hemostasis, and antibacterial, antioxidant, anti-inflammatory properties. In addition, the three-dimensional network system mimics the microarchitecture of the native ECM, thus providing convenience for cell survival, transplantation, differentiation, and endogenous regeneration. All these desired properties endow the HQCS-SP hydrogel with great application potential in wound dressings and tissue regeneration.

## 2. Materials and Methods

### 2.1. Materials

Chitosan (CS, deacetylated ~85%, 100–200 mPa·s, Mv = 125–265 kDa), 1,1-diphenyl-2-picrylhydrazyl (DPPH) free radical (97%) and 2,2′-azinobis(3-ethylbenzothiazoline-6-sulfonic acid ammonium salt (ABTS) were obtained from Aladdin Reagent Inc. (Los Angels, CA, USA). Hydrocaffeic acid (HCA) and 3-(3-dimethylaminopropyl)-1-ethylcarbondiimide hydrochloride (EDC·HCl) were purchased from Shanghai Adamas Reagent Co., Ltd. (Shanghai, China). Glyc-idyltrimethylammonium chloride (ca. 80% in water) (GTMAC) was purchased from TCI Development Co., Ltd. (Shanghai, China). Spirulina was purchased from Shanghai Gu-angyu Biotechnology Co., Ltd. (Shanghai, China). 2′,7′-dichlorofluorescin diacetate (DCFH-DA), 3-(4,5-dimethylthiazol-2-yl)-2,5-diphenyltetrazolium bromide (MTT) and lipopolysaccharide (LPS) were obtained from Sigma-Aldrich (St. Louis, MO, USA). Mouse IL-6 and TNF-*α* enzyme-linked immunosorbent assay (ELISA) kits were purchased from Signalway Antibody LLC (College Park, MD, USA) (https://www.sabbiotech.com/, accessed on 21 February 2023). The NIH-3T3 cell line (BNCC100843) was obtained from BeNa Culture Collection Co., Ltd. (Xinyang, China). Other chemical reagents were analytically pure and used without further purification. 

### 2.2. Preparation of QCS-HCA (HQCS) and Spirulina Protein (SP)

First, the quaternized chitosan (QCS) was synthesized based on a previous reported method [38]. The detailed procedure is presented in Appendix A. Afterwards, the residual amino and primary hydroxyl groups in QCS reacted with carboxyl groups in HCA to produce HQCS. Specifically, a solution containing 1.0% (*w*/*v*) QCS was prepared. Then, a solution containing 1.0% (*w*/*v*) HCA and a solution containing 1.0% (*w*/*v*) EDC (dissolved in DDW and ethanol (1:1, *v*/*v*)) were added dropwise into the above solution. Finally, the pH of the reaction mixture was adjusted to 5 with vigorously stirring at room temperature for 12 h. After that, the resulting solution was dialyzed (MWCO of membrane: 8000–14,000) for one week. The final product was obtained by freeze-drying (Scientz-10N, Ningbo Scientz Biotechnology Co., Ltd., Ningbo, China) for five days and stored at 2–8 °C, which was then characterized by Fourier transform infrared (FTIR, Nicolet 6700, Thermo Fisher Technology Co., Ltd., Waltham, MA, USA), Nuclear magnetic resonance (NMR, Avance III. HD 400 MHz, Bruker, Switzerland), and UV-vis spectroscopy (TU-1950, Beijing General Analytical Instrument, Beijing, China) (Appendix A). SP with a protein content of 82.5% (*w*/*v*) was prepared by a freeze–thawing method described in our previous work [39].

### 2.3. Formation of Complex Hydrogels

For a typical coordination compound hydrogel, HQCS and SP were entirely dissolved in DDW, respectively. Then, 0.1 M FeCl_3_ solution (Fe^3+^: HQCS = 1:3, molar ratio) was added to the HQCS/SP solution under agitation at room temperature, and the hydrogel (HQCS-SP) was ultimately formed. The obtained hydrogel was then characterized by FTIR, NMR and scanning electron microscope (SEM, S-4800, Hitachi Corporation, Tokyo, Japan).

### 2.4. Gelation Behavior, Self-Healing Behavior and Injectability of the Hydrogels

The rheology of hydrogels was evaluated with TA Instruments (Discovery DHR-2, New Castle, DE, USA). The energy storage modulus (*G*′) and loss modulus (*G*″) were studied by placing the mixture between parallel plates with a diameter of 8 mm and a spacing of 1 mm. The details of this are presented in the Appendix A.

The self-healing capacity of hydrogels was evaluated through a rheological recovery test and macroscopic autonomous healing. Then, the injectability was tested by a manual push-out test. The HQCS/SP and FeCl_3_ were added to different channels of the binocular syringe, respectively, while pushing the pressure lever so that it mixed rapidly in a tee tube and then injected through a silicone hose.

### 2.5. In Vitro Antibacterial Properties, Antioxidant Activity, Cytotoxicity Assay and Cell Migration Experiment of Hydrogels 

In this study, the antibacterial properties of hydrogels against *Staphylococcus aureus* (*S. aureus*) and *Escherichia coli* (*E. coli*) were investigated by plate and inhibition zone methods. In addition, the morphology of bacteria treated with different treatments was observed by SEM. The detailed information are presented in the Appendix A.

The defense against oxidative stress was mainly achieved by the scavenging of free radicals. The antioxidant activity of the hydrogels was evaluated by scavenging the ABTS·^+^, DPPH, and ·OH. The detailed description is presented in the Appendix A.

The in vitro cytotoxicity assay of hydrogels was determined by MTT assay. A cellular oxidative stress model was established using a H_2_O_2_-induced NIH-3T3 cell, which was then applied to evaluate the ability of hydrogels to eliminate intracellular ROS. The details are presented in the Appendix A.

The effect of hydrogels on the healing process in vitro was studied via cell migration assays. The details are presented in the Appendix A.

### 2.6. In Vitro Anti-Inflammatory Test

The anti-inflammatory ability of the hydrogel in vitro was assessed by LPS stimulation. In detail, RAW 264.7 cells (~1 × 10^5^ cells/mL) were seeded in a 96-well plate (100 μL) and cultured for 24 h. Afterwards, 100 μL of medium containing different concentrations of HQCS and SP was added to the cells treated with or without LPS (1 μg/mL) stimulation for 24 h. Subsequently, the amount of pro-inflammatory factors (TNF-*α* and IL-6) in the supernatant was measured by the ELISA kit. The anti-inflammatory test was repeated three times.

### 2.7. In Vitro Hemostatic Test, Procoagulant Test and Hemocompatibility Assay of Hydrogels

The hemostatic properties of hydrogels were evaluated using a mouse liver hemostatic model. The procoagulant properties of hydrogels were evaluated by the coagulation index (BCI) [40]. The hemocompatibility of the hydrogel was assessed by mouse red blood cell (RBC) suspension. The details were presented in Appendix A.

### 2.8. Wound Healing of Hydrogels on Full-Layer Skin Defect Models and Histological Analysis

The effect of hydrogels on wound healing was evaluated using a full-layer mouse skin model. Female Balb/c mice (17–20 g) were used as experimental subjects according to a previous report [41]. All experimental protocols in this study were approved by the Ethics Committee of the Center for Animal Experimentation of Jiangnan University. All animal laboratory operations were performed in accordance with the National Research Council’s Guide for the Care and Use of Laboratory Animals. Wound tissue was collected and subjected to hematoxylin-ebony (H&E) and Masson trichromatic staining. Accordingly, the granulation tissue thickness and proportion of collagen deposition area were calculated by Aipathwell softwell (Servicebio, Wuhan, China). The tissue homogenizations were analyzed for levels of inflammatory factors IL-6 and IL-10. The details are presented in the Appendix A.

### 2.9. Statistical Analyses

All the experimental data were statistically analyzed as a mean ± standard deviation (SD). The data was analyzed by one-way ANOVA followed by Tukey’s test (GraphPad Prism 8). The difference was considered statistically significant when *p* < 0.05.

## 3. Results and Discussion 

### 3.1. Preparation and Characterization of Hydrogels

Modified CS (HQCS) was synthesized by grafting CS with glycidyl trimethylammonium chloride (GTMAC) and HCA. QCS was first synthesized by introducing quaternary ammonium salt groups with sizeable steric resistance and strong hydration ability to CS to improve its water solubility. HCA was then introduced into QCS using the classical EDC amidation reaction (Figure 1a). The degree of CS grafts quaternary ammonium salt and catechol groups was about 67.8% and 22.3%, respectively, as confirmed by proton nuclear magnetic resonance spectroscopy (^1^H NMR), UV-vis spectroscopy and FTIR (Appendix A). Afterwards, the hydrogel was fabricated by mixing HQCS, SP, and FeCl_3_ solution for several minutes (Figure 2a). In the FTIR spectrum, the vicinity of 1710 cm^−1^ represents the C=O telescopic vibration peak. The N-H bending peak and C-N stretching peak near 1635 and 1556 cm^−1^ shift to high wavenumbers due to hydrogen bonding. The SEM image and the peaks in FTIR confirmed the formation of hydrogel (Figure 2b,c).

Subsequently, the properties of the hydrogel were characterized. As shown in Figure 2d, within the frequency range of 0.1–100 rad·s^−1^, the storage modulus (*G*′) of hydrogel consistently surpassed the loss modulus (*G*″), indicating that an elastic network is formed in the hydrogel. Hydrogels used as wound excipients should be able to effectively fill notches of any shape and possess self-healing properties. The viscosity of hydrogels decreases with increasing shear rate (Figure 2e), demonstrating the shear-dilution capacity of the hydrogel. At the same time, the hydrogel can be consistently injected through a syringe to form the word “JNU” while maintaining a stable gel state upon the removal of shear forces (Figure 2e inset).

Hydrogels with self-healing properties could extend its functional lifespan. Rheological recovery and macroscopic self-healing assessments were conducted to evaluate the self-healing potential of the synthesized hydrogels. Strain scans of hydrogels reveal that *G*′ and *G*″ intersect at strains around 700.0%, representing the threshold at which the hydrogel network collapses (Figure 2f); *G*′ then sharply decreases as the strain surpasses the critical strain. Following this, a series of alternating strain tests were carried out to test the intrinsic healing properties of hydrogel (Figure 2g). Upon applying a high dynamic strain (700.0%), the *G*′ value dropped from ~2100 Pa to ~532 Pa, falling below *G*″, which signified the disruption of the hydrogel network. When a lower strain (1.0%) was applied, the *G*′ and *G*″ values could revert to their initial levels even after three alternating repeat cycles, suggesting the hydrogel exhibits autonomous healing behavior. A macroscopic healing examination was subsequently performed to assess spontaneous healing capacity of hydrogel (Figure 2h). A cylindrical hydrogel sample was bisected, and the halves were rejoined and allowed to recover at room temperature without external intervention. After one hour, the HQCS-SP hydrogel with circular defects could fully heal at ambient temperature. These findings demonstrated that HQCS-SP hydrogels possess robust healing capabilities, which can be attributed to the intermolecular and intramolecular hydrogen bonding in HQCS-SP and the trivalent coordination between catechol groups and Fe^3+^ in HQCS-SP.

### 3.2. Antibacterial Properties of Hydrogels

Bacterial infection serves as a crucial factor impeding wound healing in clinical settings [37]. Consequently, we assessed the antimicrobial efficacy of the hydrogel against typical Gram-positive bacteria *S. aureus* and Gram-negative bacterium *E. coli* using plate count methods, and PBS employed as the control group (Figure 3a–d). Taking the control group as a reference, the killing rates of the target bacteria *E. coli* and *S. aureus* in the HQCS experimental group reached 89.9 ± 3.2% and 94.2 ± 1.6%, respectively. HQCS-SP group, on the other hand, had achieved an almost 100% killing ratio. To further intuitively evaluate the bacteriostatic ability of hydrogels, the excellent bacteriostatic properties of hydrogel could be revealed from the results of the bacteriostatic circles of each group in Appendix A. HQCS-SP inhibited the growth of target bacteria on both *S. aureus* and *E. coli* plates, resulting in the emergence of inhibition zones. All these results demonstrated that the hydrogel possessed good antibacterial activity. 

In addition, the morphology of the bacteria was characterized by SEM (Figure 3a,c). It showed that, compared with the conventional morphology of the control group, the target bacterial morphology produced different degrees of wrinkles and collapsed for the experimental groups. This may be attributed to the following points. Firstly, the highly charged polycationic structure possessed by the functionalized CS-based hydrogel interacts strongly with the negatively charged phospholipid constituents of the bacterial membrane. This interaction prompts the bacterial structure to disintegrate and constrict due to membrane impairment and the subsequent release of contents, ultimately leading to bacterial death [42]. Additionally, the adhesion of the hydrogel can effectively capture bacteria and stick them to their 3D network structure, thereby achieving the purpose of growth inhibition and targeted sterilization of bacteria. 

### 3.3. Biocompatibility and Cell Migration Ability of Hydrogels

Cytotoxicity is an important indicator that should be considered when evaluating the safety of biomedical materials. As shown in Figure 4a, the live/dead cell staining of HQCS- and SP-treated cells within 48 h showed the normal growth state of the cells. In addition, the MTT test results of HQCS, SP, and HQCS-SP on NIH-3T3 cells all showed low toxicity and even promoted cell proliferation (Figure 4b and Appendix A). These results demonstrated the excellent biocompatibility of HQCS-SP-based hydrogels, which can be attributed to the natural origin and nontoxic properties of CS and SP.

Cell migration benefits fibroblasts with collagenase in wound healing and tissue reconstruction after repair. The cell migration assays in vitro were then performed. The results were shown in Figure 4c. Compared with the control group, cells in each experimental group could migrate to the center of the scratch faster to cover the scratch wound, indicating that the hydrogel could promote the migration of fibroblasts. Relative to the control group, the wound healing rates of the HQCS and SP groups were increased by 12.2% and 13.8% at 12 h, respectively. The wound healing rates of the HQCS-SP groups increased by 28.3% at 12 h. In particular, HQCS-SP could accelerate cell migration more significantly, and showed a significant difference at 48 h. The wound healing rate reached 97.8 ± 0.2% at 48 h; the scratch wound is almost completely covered, while the healing rate of the control group was only 76.6 ± 0.6% (Figure 4d). Based on the healing rate data of HQCS and SP, the excellence of HQCS-SP may be attributed to the active peptides from SP, which provided the appropriate kinetics for cell migration. These results demonstrated that the HQCS-SP hydrogel had excellent wound closure ability in vitro. 

### 3.4. Antioxidant Activity and Anti-Inflammatory Properties of Hydrogels

Excessive ROS generated at the wound site will cause oxidative stress damage to normal tissues and hinder wound healing [43]. Here, we studied the antioxidant capacity of hydrogel through ABTS·^+^, DPPH, and ·OH scavenging experiments. As shown in Figure 5a–f, the HQCS-SP has an excellent ability to remove ABTS·^+^, DPPH, and ·OH. Moreover, the scavenging activity of HQCS-SP against free radicals increased with its concentration. Additionally, HQCS shows a more vital ability to scavenge free radicals than SP (Appendix A), which may be attributed to the existence of numerous phenolic hydroxyl groups in HQCS. It is reported that phenolic hydroxyl plays a role in the antioxidant capacity of polyphenols through its hydrogen atom transfer and single electron transfer capability. Therefore, HQCS-SP with great antioxidant effect may be attributed to HQCS. 

Afterward, the antioxidant capacity of hydrogels was explored at the cellular level using a fluorescent probe, and H_2_O_2_ was utilized to create oxidative stress (Figure 5g–i). Figure 5g indicated that HQCS-SP can effectively remove intracellular ROS in a H_2_O_2_ induced NIH-3T3 cells model. Therefore, this study provided an effective strategy to combat the oxidative stress damage induced by ·OH. When ROS was present in cells, ROS oxidized DCFH to DCF with fluorescent emission, and the fluorescence intensity of DCF was detected using flow cytometry to explore the activity level of ROS in cells (Figure 5h) [44]. As shown in Figure 5i, the intra-envelope DCF fluorescence intensity of normal cells was about 1490 ± 22, while the intracellular DCF fluorescence intensity of the H_2_O_2_ damage group reached 5282 ± 758, indicating that H_2_O_2_ induced the production of a large amount of ROS in cells. When cells were pretreated with HQCS and HQCS-SP, the intracellular fluorescence intensity decreased to 2655 ± 146 and 2315 ± 162, respectively. The results showed that HQCS-SP could effectively inhibit oxidative stress in cells. It is worth noting that the intracellular fluorescence intensity decreased to 1822 ± 284 when the cells pretreated with SP, which suggested that SP shows efficacy in combating oxidative stress. These results demonstrated that HQCS-SP had a defensive effect against oxidative stress [45,46]. 

In addition, wound tissue, due to bacterial infection and excess ROS, will cause an inflammatory response and hinder wound healing [47]. Pro-inflammatory cytokines, particularly interleukin-6 (IL-6) and tumor necrosis factor-*α* (TNF-*α*), exhibit increased expression during the inflammation phase of wound healing. HQCS, SP and HQCS-SP exhibited low toxicity to Raw 264.7 cells (Appendix A). As illustrated in Figure 5j,k, in the LPS-induced RAW264.7 macrophage inflammation model, both HQCS and SP, could reduce the levels of IL-6 and TNF-*α*. Additionally, HQCS-SP significantly suppressed IL-6 and TNF-*α* expression in comparison to the negative control group. This effect could be attributed to SP, which can inhibit nuclear factor κB (NF-κB) activation by preventing the degradation of cytosolic IκB-α in LPS-stimulated RAW264.7 macrophages [35]. Such findings offer valuable support for subsequent research aimed at enhancing wound healing.

### 3.5. In Vitro Blood Coagulation and In Vivo Hemostasis of Hydrogels

As a wound dressing hydrogel, it should have good blood compatibility; that is, it should satisfy the hemolysis ratio of the material by at least less than 5.0% (ISO 10993-4) [40]. The hemolytic activity assay evaluated the hemocompatibility of the hydrogels. As shown in Figure 6a, Triton X-100 was used as a negative control. Due to the rupture of erythrocytes in the blood, the amount of free hemoglobin in the erythrocytes decreased, causing hemolysis, and the supernatant showed a uniform bright red solution. The supernatant of the hydrogel dispersion group was nearly colorless, resembling the negative control PBS solution. The hemolysis rate of the hydrogels in the HQCS-SP group reached 2.0 ± 0.1%, while the HQCS groups were less than 1.5%, as low as 1.3 ± 0.1%. It showed that HQCS-SP and HQCS hydrogels had good blood compatibility [48].

Bleeding is one of the main complications after trauma, and bleeding exceeding 20.0% of the systemic blood volume can lead to shock or even death. As a wound dressing, coagulation performance and hemostasis are two of their essential indicators [49]. First, the coagulation performance of these hydrogels was assessed through the in vitro blood coagulation index (BCI). The coagulation promotion effect can be seen intuitively from Figure 6b, the lower the BCI value, the faster the coagulation rate. As shown in Figure 6c, the BCI decreased over time, indicating that the blood was gradually coagulating. It is worth noting that the BCI value of the hydrogel was lower than that of the control group and the Hydrosorb group at different time points. The BCI of the Hydrosorb group changed from 73.8% to 51.8% with the time increase from 30 s to 120 s. While the BCI of the HQCS-SP group decreased from 43.2% to 19.4%, the BCI of the HQCS group decreased from 29.3% to 12.2%, indicating that the hydrogel had an immediate procoagulant effect. 

Considering the in vitro coagulation properties, we hypothesize that the hydrogels also possess outstanding hemostatic capabilities. The hemostasis capability of hydrogel was then evaluated using the Balb/c male mouse liver tear. The schematic diagram of the mice liver bleeding model and hydrogel hemostasis process were shown in Figure 6d. Hydrosorb and hydrogels could stanch bleeding by adhering to the wound bed and stimulating the release of coagulation factors [50]. The hemostasis result was shown in Figure 6e; the control group left a giant bloodstain, while the Hydrosorb and hydrogel groups both left smaller bloodstains. The quantitative result was shown in Figure 6f,g, the blood loss and hemostasis time of the Hydrosorb, HQCS-SP, and HQCS groups were significantly reduced compared with the control group. The bleeding volume of the HQCS-SP group (27 ± 2 mg) was reduced by 82.0% compared with the control group and by 39% compared with the commercial dressing Hydrosorb group. The change in hemostasis time showed a significant difference, which was reduced to 41 ± 4 s in the Hydrosorb group compared with 157 ± 2 s in the control group. The hemostasis time of HQCS-SP and HQCS was reduced to 22 ± 1 s and 15 ± 1 s, respectively, at least 5~6-fold reduction. It illustrated that hydrogels had good wound closure and hemostasis effects [51]. It gives credit to the fact that when the hydrogel is in contact with the bleeding wound bed tissue and rapidly gelled, the formed hydrogel has strong adhesion, which can achieve rapid hemostasis by closing the wound and forming a physical barrier for the hemostasis effect. The excellent swelling properties of hydrogel enable effective absorption of serum water, ultimately concentrating components such as blood cells and coagulation factors. In addition, CS possesses innate hemostatic properties. The modified positively charged quaternary ammonium salt groups may form electrostatic interactions with blood cells, leading to the adsorption and accumulation of blood cells. SP has a specific stabilizing effect on erythrocytes and platelets, thus giving the hydrogel better coagulation ability [52].

### 3.6. In Vivo Wound Healing Performance of Hydrogels

Subsequently, the wound-healing efficacy of these hydrogels was assessed in the mouse full-thickness skin defect model. In Figure 7a, the wound progressively closed over time, with the healing effect of the experimental group at various time points being significantly greater than that of the control group. During the initial three days of full-thickness skin injury, basal cells at the wound periphery started proliferating and migrating beneath the clot towards the wound center, forming a monolayer of epithelium that covers the granulation tissue surface. Upon encountering contact inhibition, these basal cells ceased migration and differentiated into squamous epithelium. The Hydrosorb group exhibited a 20.1% increase compared to the control group (Figure 7b). In contrast, the wound healing rates of the HQCS and HQCS-SP groups were faster than that of the Hydrosorb group, reaching 55.8 ± 1.7% and 63.4 ± 2.1%. Wounds involving tissue necrosis and blood vessel rupture induce an inflammatory response, vascular congestion, and inflammatory cell exudation, resulting in redness and swelling. Blood and fibrinogen seeping from the wound coagulate into clumps, forming scabs that safeguard the wound. As the muscle fibroblasts of the wound skin and subcutaneous tissues proliferated, the wound was pulled, and the edge of the wound moved toward the center, which made the wound shrink [53]. On day 7, the healing rate of the Hydrosorb group reached 65.0 ± 3.8%, which was 23.8% faster than that of the blank group. In addition, the wounds covered by HQCS-SP hydrogels had better healing performance, the redness and swelling of new tissues were reduced, and the healing rate reached 81.9 ± 2.0%. In the later stages of healing, the blood scabs fall off, the wound tended to heal completely, and eventually, different degrees of scarring were formed. On day 14, the healing rate in the control group was 77.8 ± 1.3%, the healing rate in the Hydrosorb group was 87.9 ± 2.4%, and the wound was still accompanied by redness and blood scabs. The sample group ultimately completed the scab removal behavior from the 11th day, and the wound healing rate in the HQCS group reached 96.5 ± 1.0%, which was close to complete closure (Figure 7c). While the HQCS-SP groups were healed entirely, and the wound was smooth and shiny. In summary, the above results showed that hydrogels have good wound healing ability, mainly due to the following aspects. Firstly, the powerful adhesion assures close contact between those hydrogels and surrounding tissues while isolating the wound from the detrimental external environment. Exceptional water absorption aids in the absorption of wound exudate, while the high antibacterial activity of hydrogel safeguards the wound against bacterial infections during the healing process. Secondly, the catechol group and natural product SP present in the hydrogel boost cell movement and adherence, enhance organism compatibility, intelligently modulate the ECM protein signaling pathway, and establish an optimal ECM microenvironment conducive to wound healing.

Prolongation of the inflammatory phase leads to the reduction in growth factor production, as well as a surfeit of inflammatory cytokines and overexpression of matrix metalloproteinases, which retards wound healing [54]. Quantitative assessments based on the pro-inflammatory factor IL-6 and the anti-inflammatory factor IL-10 revealed that inflammation was most pronounced in the control group (Figure 7d,e). In contrast, wound tissues treated with HQCS and HQCS-SP groups displayed comparatively milder inflammation. Simultaneously, the IL-10 levels were elevated by 204.6 pg/mL and 240.5 pg/mL compared to the control group. By day 7, inflammation in each group had subsided, but the wound area in the control group remained at a higher-than-normal level of inflammation. Conversely, the Hydrosorb group demonstrated a 29.0% reduction in inflammation compared to the control group, and the hydrogel group, especially HQCS-SP, was weakened by 55.6%. This may be attributed to the sustained release of active inflammatory factors from the HQCS-SP hydrogel, which stimulates acute and chronic inflammatory cells, including macrophages, neutrophils, and lymphocytes, to migrate to the wound region to reestablish microenvironmental homeostasis. From day 7 to day 14, as granulation tissue grows and the epidermis and dermis form, the inflammatory response in the healing tissue progressively diminishes.

Furthermore, the healing effect of hydrogel dressings was assessed from a histological point of view. The regenerated skin tissue collected on days 3, 7, and 14 was stained with hematoxylin, ebony red (H&E), and Masson (Figure 8). We then analyzed the overall histological morphology of the granulation tissue and collagen regeneration of the wound bed. In the inflammatory stage, more connective tissue necrosis could be seen in the granulation tissue of the control group, and the arrangement was irregular. In comparison with the control group, the dermis and subcutaneous hyperplasia repair of skin tissue in the Hydrosorb and experimental groups could be seen, which had a variety of functions such as connection, support, protection, storage of nutrients, and material transportation (Figure 8a). The granulation tissue of the HQCS-SP group (745 ± 34 μm) was about 165 μm thicker than that of the Hydrosorb group (580 ± 34 μm) and 1.5 times thicker than that of the control group (488 ± 12 μm) (Figure 8b). At the same time, a small number of neovascularizations emerged, accompanied by more neutrophil and macrophage infiltration. At the proliferation stage, the wound was full of granulation tissue, which was composed of newly formed capillaries and connective tissue, fibroblasts, growth factors, accumulated ECM, and other cells [55]. The formation of new capillaries provides nutrition for the growth of granulation tissue and is crucial to wound healing. On the 7th day, the control group still had more inflammatory exudates, such as eosinophils, that caused tissue damage and promoted the progress of inflammation. Additionally, collagen deposition in skin tissue was observed by Masson trichrome staining (Figure 8c). Compared with the control group and Hydrosorb group, collagen deposition was denser in the HQCS-SP group. Collagen deposition increased by 23.5% compared to the control group (Figure 8d). These results showed that HQCS-SP hydrogel dressing could significantly accelerate tissue regeneration and wound healing.

## 4. Conclusions

In conclusion, we designed and prepared a microalgae-type biofunctional hydrogel, examined its antioxidant, anti-inflammatory, antibacterial activity, and explored its potential application in full-thickness skin wound repair. The results demonstrated the HQCS-SP hydrogel with good injectability and self-healing property was successfully obtained. Furthermore, the prepared HQCS-SP hydrogel displayed excellent antibacterial, antioxidant and anti-inflammatory activity, which could effectively combat the bacterial infections, oxidative stress damage and inflammation in wounds, and accelerate the wounds repair. At the same time, the hydrogel had an outstanding cell migration ability, biocompatibility, and hemostasis capability, which could promote wound closure with high biosafety. In general, the HQCS-SP hydrogel with various desired properties promoted the wound closure and wound closure post-care efficiently, which provides a new idea and application for skin wound repair in natural state, particularly as a promising candidate material for full-thickness skin damage restoration. In addition, new possibilities are opened up for the design of composite materials with a wider range of healthcare applications, considering bio-based hydrogels made by HQCS-SP supplemented by hydrocolloids or customized nanoparticles [56,57]. Through the introduction of hydrocolloids to improve physical properties, as well as new capabilities enabled by nanoparticles, such as targeted drug release, the development of these advanced materials will enable the emergence of more effective and personalized medical solutions, opening up new opportunities in healthcare.

## Figures and Tables

**Figure 1 polymers-16-00692-f001:**
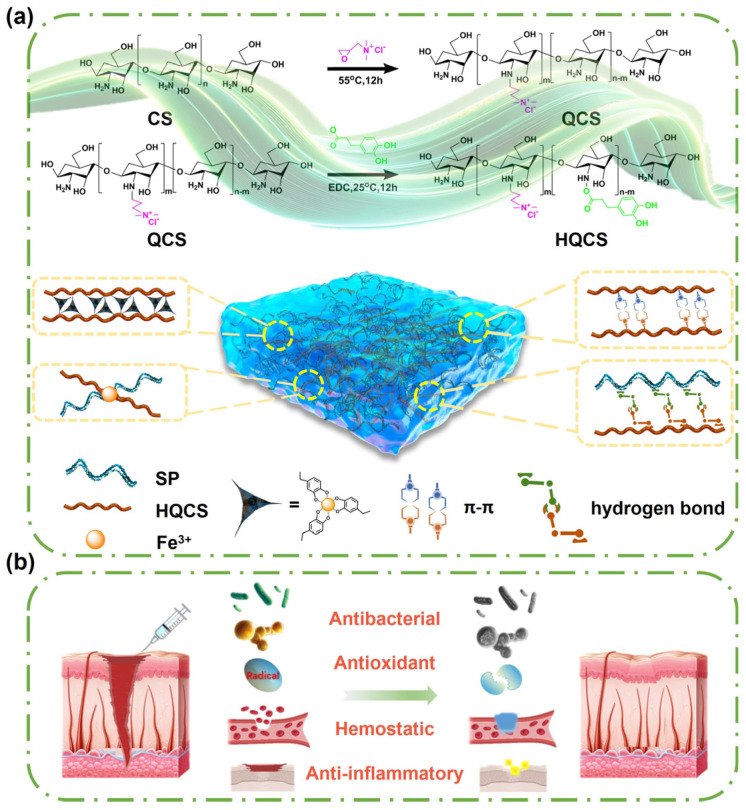
Illustration of preparation of HQCS-SP hydrogels and their application for full-thickness skin injuries wound healing. (**a**) Scheme of the preparation procedure of HQCS-SP hydrogel. (**b**) Schematic representation for the design strategy and application of biofunctional hydrogel for full-thickness skin wound repair.

**Figure 2 polymers-16-00692-f002:**
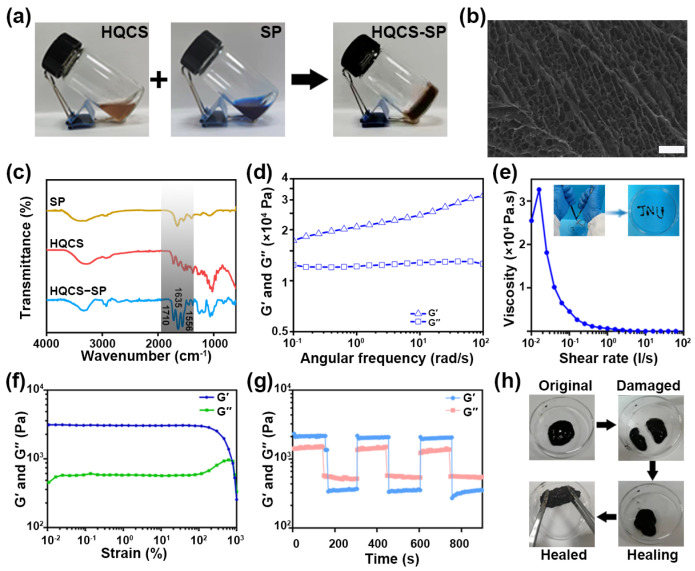
Characterization of HQCS-SP hydrogel. (**a**) Schematic diagram of crosslinking of the hydrogel. (**b**) SEM image of the hydrogel after freeze-drying. Scale bar: 100 µm. (**c**) FTIR spectroscopy of SP, HQCS and HQCS-SP. (**d**) Rheological behaviors of the hydrogel. (**e**) Shear-thinning properties of the hydrogel. Inset: Presentation of the injectability of the hydrogel. (**f**) Strain sweeps of the hydrogel with the strain ranging from 0.01% to 1000.0% (1 Hz). (**g**) Rheological behaviors of the hydrogel with alternate strains switched from 1.0% to 700.0% for three cycles. (**h**) Photographs of the macroscopic healing capacities of the hydrogel.

**Figure 3 polymers-16-00692-f003:**
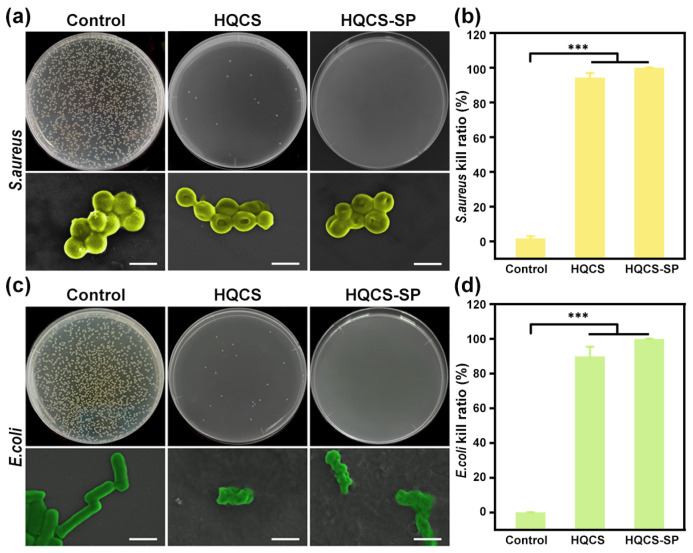
Antibacterial activity of hydrogels against *S. aureus* and *E. coli*. (**a**) Plate counting effects of hydrogel precursor solutions on target bacteria. The yellow false-colored part shows *S. aureus* in SEM renderings. Scar bar: 1 µm. (**b**) Kill ratio of *S. aureus*. (**c**) The green false-colored part shows *E. coli* in SEM renderings. Scar bar: 1 µm. (**d**) Kill ratio of *E. coli*. (*** *p* < 0.001).

**Figure 4 polymers-16-00692-f004:**
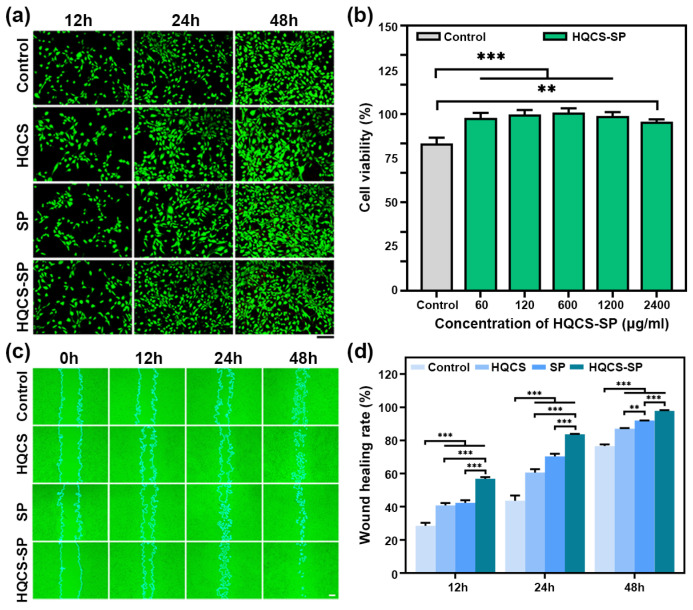
Biocompatibility and cell migration ability of hydrogels. (**a**) Live/dead staining of NIH-3T3 cells after treatment with HQCS-SP for 12, 24, and 48 h, respectively. Scale bar: 250 µm. (**b**) Cell viability of HQCS-SP at different concentrations. (**c**) Representative photographs of cell scratches at 0 h, 12 h, 24 h, and 48 h after treatment with culture medium, different concentrations of HQCS, SP, and HQCS-SP. Scale bar: 200 µm. (**d**) The wound healing rate of cells treated with HQCS, SP and HQCS-SP. (** *p* < 0.01, *** *p* < 0.001).

**Figure 5 polymers-16-00692-f005:**
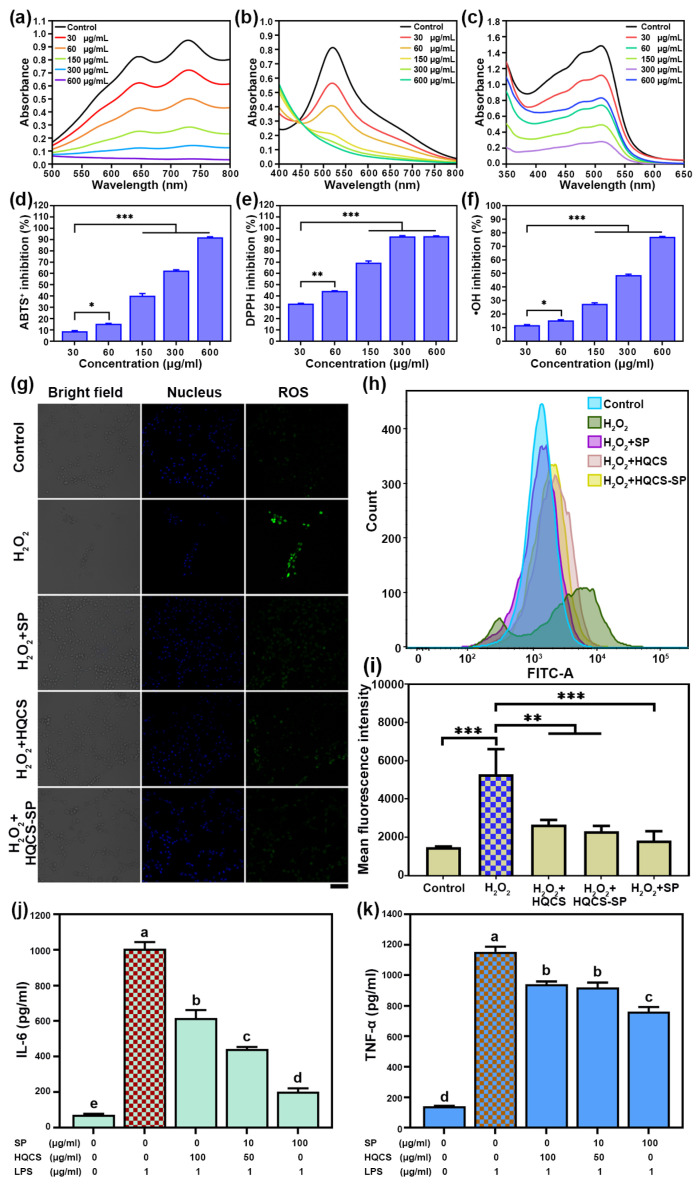
Antioxidant activity and anti-inflammatory properties of hydrogels. (**a**–**f**) The antioxidant capacity of different concentrations of HQCS-SP to scavenge ABTS·^+^, DPPH, and ·OH. (**g**) Confocal fluorescence images of NIH-3T3 cells stained with DCFH-DA and Hoechst 33342 after different treatments, Scale bar: 100 µm; (**h**) Effects of DCFH-DA fluorescence intensity. (**i**) Average fluorescence intensity of DCF in cells. (**j**,**k**) The expression of pro-inflammatory factors (IL-6, TNF-*α*) in LPS-activated RAW 264.7 cells with different treatments. (* *p* < 0.05, ** *p* < 0.01, *** *p* < 0.001, different low case letters above columns indicate statistical differences at *p* < 0.05).

**Figure 6 polymers-16-00692-f006:**
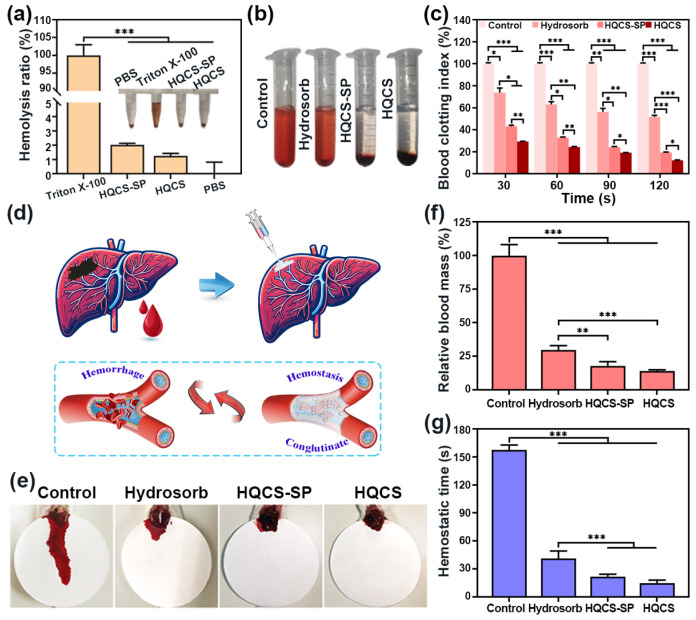
Coagulation and hemostatic properties of hydrogels. (**a**) Hemolysis ratio (*n* = 3). (**b**) Representative images of the supernatants in different groups after the blood-clotting test. (**c**) Changes in blood clotting index (BCI) over time (*n* = 3). (**d**) Schematic diagram of the mice liver hemorrhage model and hydrogel hemostasis process. (**e**) Bleeding circumstances. (**f**) Relative blood loss (*n* = 5). (**g**) Hemostasis time (*n* = 5). (* *p* < 0.05, ** *p* < 0.01, *** *p* < 0.001).

**Figure 7 polymers-16-00692-f007:**
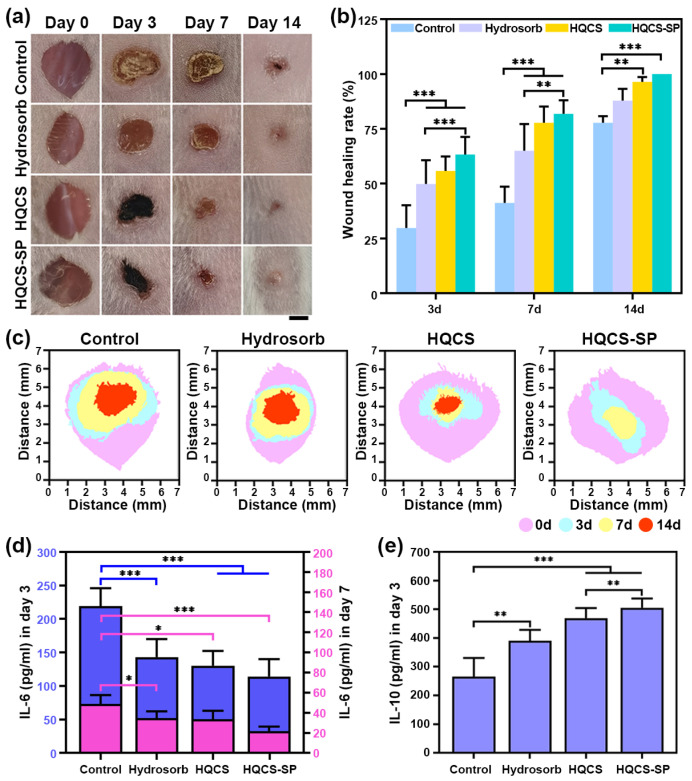
A full-layer mouse skin defect model that assesses wound healing. (**a**) Representative photograph of wound healing under different treatment modalities. Scale bar: 2 mm. (**b**) Quantification of wound closure rate. (**c**) Graph of the evolution of wound area by the group over time. (**d**) IL-6 and (**e**) IL-10 content in the wound sites of various groups on different days by ELISA assay. (* *p* < 0.05, ** *p* < 0.01, *** *p* < 0.001).

**Figure 8 polymers-16-00692-f008:**
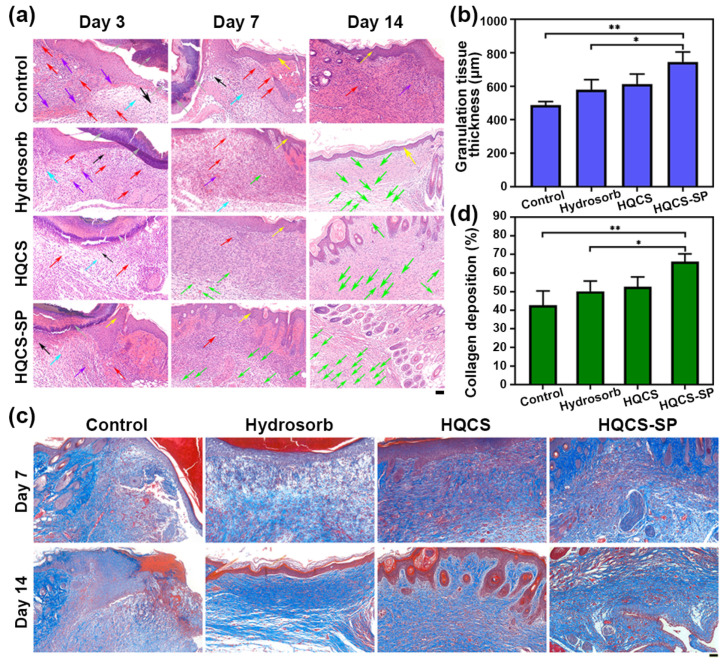
Histological analysis of wound regeneration. (**a**) H&E staining. Blue arrows indicate connective tissue; green arrows indicate neovascularization; purple arrows indicate the site of bleeding; red arrows indicate inflammatory cells; yellow arrows indicate the epidermal spines; gray arrows indicate inflammatory exudates; black arrows indicate connective tissue necrosis. Scale bar: 50 µm. (**b**) Granulation tissue thickness in each group on day 14. (**c**) Masson staining. Scale bar: 50 µm. (**d**) Proportion of collagen deposition area in each group on day 14. (* *p* < 0.05, ** *p* < 0.01).

## Data Availability

Data is contained within the article.

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
