# Peer review of "Compound Microalgae-Type Biofunctional Hydrogel for Wound Repair during Full-Thickness Skin Injuries"

_polymers, 2024, doi:10.3390/polym16050692_

Round 1

Reviewer 1 Report

Comments and Suggestions for Authors

The manuscript titled “Compound microalgae-type biofunctional hydrogel for wound repair during full-thickness skin injuries” by Mao, Y.; et al. is a scientific work where the authors fully characterize the performance of hydrogels made of chitosan crosslinked with biosourced spirulina protein in presence of ferric ions. Many complementary techniques were used in this research like fourier transform infrared spectroscopy (FTIR), scanning electron microscopy (SEM), rheology measurements, nuclear magnetic resonance (NMR), antibacterial citotoxicity, hemocompatibility and anti-inflamatory assays being the wound healing capacity interrogated at different scale times (from 0 to 14 days). This is a well-designed work and it could be interesting for a certain target audience. Furthermore, the manuscript is generally well-written.

However, it exists some points that need to be addressed (please, see them below detailed point-by-point). The most relevant outcomes remarked by the authors can contribute in the growth of many fields like the development of more efficient materials for wound healing and tissue engineering. For this reason, I will recommend the present scientific manuscript for further publication in the Polymers once all the below described suggestions will be properly fixed.

Here, there exists some points that must be covered in order to improve the scientific quality of the manuscript paper:

1) KEYWORDS. (OPTIONAL) The authors should consider to add the chemistry compounds which are forming the tested hydrogels (chitosan and spirulina protein) in the keyword list.

2) INTRODUCTION. “Various factos such as accidents, surgeries (…) chronic skin injuries (…) and society” (lines 26-28). Here, some further quantitative details about the existing worldwide skin injury burdens should be furnished [1].

[1] Flohr, C.; et al. Putting the burden of skin diseases on the global map. Br. J. Dermatol. 2021, 184, 189-190. https://doi.org/10.1111/bjd.19704.

3) “Among them, chitosan (CS) (…)” (lines 45-46). Why did the authors define the chitosan abbreviation as “CS” but the crosslinked modified chitosan as “QH” (please, see the line 12 in the Abstract section)?

4) Figure 1 (line 87). The lettering of this figure is slightly blurry. Maybe this issue comes from the PDF conversion step. In this case, the authors should contact to the Assistant Editor.

5) MATERIALS AND METHODS. “ Specifically, a solution containing 1% (w/v) QCS (…)1% (w/v) HCA (…)82.5% (w/v) was prepared by a freeze-thawing method described in our previous work” (lines 112-123). Please, the authors should homogenize the significant figures displayed for the same physical property. This comment should be taken into account for the rest of the main manuscript body text.

6) “The final product (…) characterized by FTIR (…) NMR (…)” (lines 117-120). The full-name of all the used techniques should be furnished. Then, the abbreviations should be placed between brackets. This point should be covered for the rest of this section (e.g. for scanning electron microscopy in the line number 128).

7) “Then, 0.1 M FeCl3 solution (Fe3+: QH = 1:3)” (line 126). Are the authors referring “w/v” to this ratio?

8) “2.4 Gelation behaviour, self-healing behavior and injectability of the hydrogels” (lines 130-139). The equations according to the obtention of the energy storage modulus and loss modulus should be furnished. This information will significantly aid to the potential readers to better undertand what are the key samples parameters which more impacts on these properties.

9) “In addition, the morphology of bacteria treated with different treatments was observed by SEM. The detailed information was presented in SI, materials and methods section” (lines 144-146). What was the acceleration voltage (in keV) used by the authors? (This information lacks in the SI section. Then, did the authors consider that the gold sputtered on the tested samples could lead a limitation during the SEM data interpretation? A brief statement should be provided in this remark.

10) RESULTS AND DISCUSSION. Figure 3, panels b and c (line 246). ANOVA combined with ad-hoc Tukey’s test should be devoted in these assessed conditions. Similar comment in the Fig. 4, panel b (line 286) and Fig. 5 panels d-f (line304).

11) “3.6. In vivo wound healing performance of hydrogels” (lines 395-487). The authors monitored the hydrogel response for 14 days (0, 3, 7 and 14 days, respectively). Did the authors also assessed the biodegradability (e.g. by scanning electron microscopy images) of the formulated hydrogels? Did the authors observe some loss of their architecture during the exposition time?

12) CONCLUSIONS. This section clearly states the most relevant outcomes found in this work. The authors should add a brief statement about the future action lines to pursue this research. In this context, it may be opportune to highlight the possibility to expand the promising perspectives of crosslinked chitosan hydrogels with spirulina with other formulations like hydrocolloids [2] or customized nanoparticles [3] to design materials with more broad spectrum of healthcare applications. Finally, the bibliography references are in the proper format of Polymers journal.

[2] Pele, K.G.; et al. Hydrocolloids of Egg White and Gelatin as a Platform for Hydrogel-Based Tissue Engineering. Gels 2023, 9, 505. https://doi.org/10.3390/gels9060505.

[3] Capanema, N.S.V.; et al. Nanosilver-Functionalized Hybrid Hydrogels of Carboxymethyl Cellulose/Poly(Vinyl Alcohol) with Antibacterial Activity for Prevention and Therapy of Infections of Diabetic Chronic Wounds. Polymers 2023, 15, 4542. https://doi.org/10.3390/polym15234542.

Comments on the Quality of English Language

The manuscript is well-written as above described albeit it may be desirable if the authors could recheck it in order to polish some final details susceptible to be improved. 

Author Response

Dear Reviewer:

Thank you very much for your comments and professional advice concerning our manuscript entitled “Compound microalgae-type biofunctional hydrogel for wound repair during full-thickness skin injuries” (ID: polymers-2874209). These comments are all valuable and very helpful for improving our paper. We have studied comments carefully and made corrections which we hope meet with approval.

Reviewer 2 Report

Comments and Suggestions for Authors

I had thoroughly studied the manuscript “Compound microalgae-type biofunctional hydrogel for wound repair during full-thickness skin injuries”. The author did interesting research work based on preparation of hydrogel for wound repair. The figure provided by the author is attractive and well organized. In my opinion, this research can consider for publishing in a journal of applied polymer science after addressing of the following comment. In my opinion, this research can be considered for publication in the journal of applied polymer science after addressing the following comment:.

Provide information about why you choose these two bacteria and include it in revised manuscript.

Provide information about why you choose these NIH-3T3 for cell study and include it in revised manuscript.

What is positive control for BCI measurement? Provide it and add to revised manuscript. 

Author Response

Dear Reviewer:

Thank you very much for your comments and professional advice concerning our manuscript entitled “Compound microalgae-type biofunctional hydrogel for wound repair during full-thickness skin injuries” (ID: polymers-2874209). These comments are all valuable and very helpful for revising and improving our paper, as well as the important guiding significance to our research. We have studied comments carefully and have made correction which we hope meet with approval.

Round 2

Reviewer 2 Report

Comments and Suggestions for Authors

The author responds to the comment in a satisfactory manner. Therefore, this version of the paper can be accepted by the journal.